# Prognostic Relevance of CD4^+^, CD8^+^ and FOXP3^+^ TILs in Oral Squamous Cell Carcinoma and Correlations with PD-L1 and Cancer Stem Cell Markers

**DOI:** 10.3390/biomedicines9060653

**Published:** 2021-06-08

**Authors:** Paloma Lequerica-Fernández, Julián Suárez-Canto, Tania Rodriguez-Santamarta, Juan Pablo Rodrigo, Faustino Julián Suárez-Sánchez, Verónica Blanco-Lorenzo, Francisco Domínguez-Iglesias, Juana María García-Pedrero, Juan Carlos de Vicente

**Affiliations:** 1Department of Biochemistry, Hospital Universitario Central de Asturias (HUCA), C/Carretera de Rubín, s/n, 33011 Oviedo, Spain; palomalequerica@gmail.com (P.L.-F.); faustinosuarezsanchez@gmail.com (F.J.S.-S.); fdoig59@yahoo.es (F.D.-I.); 2Instituto de Investigación Sanitaria del Principado de Asturias (ISPA), Instituto Universitario de Oncología del Principado de Asturias (IUOPA), Universidad de Oviedo, C/Carretera de Rubín, s/n, 33011 Oviedo, Spain; taniasantamarta@gmail.com (T.R.-S.); jprodrigot@telefonica.net (J.P.R.); 3Department of Pathology, Hospital Universitario de Cabueñes, 33394 Gijón, Spain; juliansuarezcanto@gmail.com; 4Department of Oral and Maxillofacial Surgery, Hospital Universitario Central de Asturias (HUCA), C/Carretera de Rubín, s/n, 33011 Oviedo, Spain; 5Department of Otolaryngology, Hospital Universitario Central de Asturias (HUCA), C/Carretera de Rubín, s/n, 33011 Oviedo, Spain; 6Department of Surgery, University of Oviedo, 33006 Oviedo, Spain; 7Ciber de Cancer (CIBERONC), Instituto de Salud Carlos III, Av. Monforte de Lemos, 3-5, 28029 Madrid, Spain; 8Department of Pathology, Hospital Universitario Central de Asturias (HUCA), C/Carretera de Rubín, s/n, 33011 Oviedo, Spain; veronica.blanco@sespa.es

**Keywords:** CD4, CD8, FOXP3, OCT4, Nestin, Podoplanin, TILs, oral cancer

## Abstract

This study investigates the relevance of tumor-infiltrating lymphocytes (TILs) in oral squamous cell carcinoma (OSCC). Immunohistochemical analysis of stromal/tumoral CD4^+^, CD8^+^ and FOXP3^+^ TILs is performed in 125 OSCC patients. Potential relationships with the expression of tumoral PD-L1 and cancer stem cell (CSC) markers (NANOG, SOX2, OCT4, Nestin and Podoplanin (PDPN)) are assessed. CD4^+^ and CD8^+^ TILs are significantly associated with smoking and alcohol habits. CD4^+^ and CD8^+^ TILs show an inverse relationship with NANOG and SOX2 expression, and FOXP3^+^ TILs is significantly correlated with Nestin and PDPN expression. High infiltration of CD4^+^ and CD8^+^ TILs and a high tumoral CD8^+^/FOXP3^+^ ratio are significantly associated with tumors harboring positive PD-L1 expression. Infiltration of stromal/tumoral FOXP3^+^ TILs and a low stromal CD8^+^/FOXP3^+^ ratio are significantly associated with better disease-specific survival. Multivariate analysis reveals that the stromal CD8^+^/FOXP3^+^ TILs ratio is a significant independent prognostic factor. Regarding OSCC patient survival, the CD8^+^/FOXP3^+^ TILs ratio is an independent prognostic factor. TILs may act as biomarkers and potential therapeutic targets for OSCC.

## 1. Introduction

Cancer is not only a genetic disease, it also involves an immunological basis. Oral squamous cell carcinoma (OSCC), one of the most common head and neck squamous cell carcinomas (HNSCC), is an immunosuppressive disease able to evade immune surveillance avoiding an effective immune response [1]. The tumor microenvironment (TME) refers to the cellular environment in which tumor cells or cancer stem cells (CSCs) exist [2,3]. This is the scenario where tumor-infiltrating lymphocytes (TILs) may play a role in the development of OSCC [3]. Studies to date generally agree that cytotoxic T lymphocytes (CTLs) and CD4^+^ Th1 cells are involved in effective antitumor immunity while FOXP3^+^ regulatory T cells are associated with suppression of antitumor immunity [4]. Interestingly, CD4^+^, CD8^+^ and FOXP3^+^ lymphocytes can be easily evaluated by immunohistochemistry.

Cancer cells overexpress the programmed death-ligand 1 (PD-L1), which binds to the programmed death-1 receptor (PD-1 or CD274) on activated T cells delivering an inhibitory signal to cytotoxic T lymphocytes that prevents tumor elimination from the immune system [5]. PD-L1 expression is affected by the surrounding microenvironment, and found to be generally associated with poor prognosis [6]. CSCs constitute a subpopulation of tumor cells endowed with properties such as self-renewal and long-term clonal persistence [7], maintained in specific niches within the tumor in which the interaction with the TME is crucial [8]. In turn, CSCs bear stemness properties supported by gene master regulators, such as NANOG, SOX2 and OCT4 [9], which have been implicated in oral carcinogenesis, poor differentiation and poor prognosis [10,11,12,13,14,15,16,17]. Accumulating evidence indicates that these transcription factors correlate with Nestin expression [18]. Nestin is a class VI intermediate filament protein identified as a stem cell marker during the development of the central nervous system [19]. Nestin is expressed in several tumors including OSCC, and found to regulate invasion, metastasis, cell cycle and apoptosis through regulation of cytoskeletal proteins and stemness [20]. Atsumi et al. [21] first identified Podoplanin (PDPN) as a CSC marker in squamous carcinoma cells. It has been shown that PDPN regulates stem cells in normal and tumor tissues [22], and also reported as a valuable biomarker for cancer risk assessment of malignant transformation in oral leukoplakia [23]. NANOG is also a pluripotency/CSC regulatory factor, found to be widely detected in multiple cancers, enriched in tumor cells that exhibit stem cell-like properties and with important prognostic implications in several cancer types, including OSCC [11,13]. The association between TILs and patient survival has been described in various types of cancer [1,24,25,26,27,28,29,30,31,32,33]. Specifically, in HNSCC, CD8^+^ T cells have been related to a favorable prognosis, although few studies have examined the prognostic relevance of CD8^+^ T cells’ density in OSCC [3,34,35]. The types and functional statuses of different TILs as well as their tissue localizations within the TME can determine the balance between control and promotion of cancer [35]. In a recent meta-analysis, de Ruiter et al. [36] confirmed the favorable prognostic role of CD8^+^ T cells in HNSCC, while the potential impact on OSCC prognosis remains a matter of controversy [26]. Additionally, in another recent meta-analysis, Borsetto et al. [37] investigated the prognostic role of CD4^+^ and CD8^+^ T cells in HNSCC, and found a significant reduction in the risk of death for both high CD4^+^ and CD8^+^ lymphocytes in oropharyngeal and for CD8^+^ TILs in hypopharyngeal cancers. Nevertheless, neither high CD4^+^ nor CD8^+^ lymphocytes were significantly associated with improved survival for patients with oral or laryngeal cancer, indicating that tumor location could be a more discriminating factor than TILs in terms of clinical outcome in HNSCC. In addition, the prognostic value of CD4^+^ T and FOXP3^+^ T cells requires further investigation [38,39] due to protumoral and antitumoral functions that have been postulated for FOXP3 TILs’ (Tregs’) expression in different cancers. Thus, Tregs recruitment has been associated with a worse prognosis in some tumors [28,29,30,40], while it has been unexpectedly correlated with good prognosis in head and neck, esophageal and colorectal cancers [24,39]. Therefore, the roles of CD4^+^, CD8^+^ and FOXP3^+^ T cell subpopulations in OSCC remain still inconclusive [25,26,27]. Further clarification is needed on whether lymphocyte infiltration represents a beneficial antitumor immune response or a poor prognostic factor involved in OSCC progression. The purpose of this study is to thoroughly evaluate the infiltration of CD4^+^, CD8^+^ and FOXP3^+^ TILs in a large and homogeneous cohort of OSCC specimens, their relationships with clinicopathological features, and impact on patient prognosis. Additionally, correlations between Tregs and CTLs infiltration and the expression of tumor PD-L1 and various well-established CSCs markers (i.e. NANOG, SOX2, OCT4, Nestin and PDPN) are also assessed. A better understanding of TILs as clinically relevant biomarkers in OSCC is a prerequisite to adopt new immunotherapy treatments.

## 2. Materials and methods

### 2.1. Patients and Tissue Specimens

Following approval by the Regional Ethics Committee from Principado de Asturias (date of approval 14th May 2019; approval number 136/19, for the project PI19/01255), this retrospective study was performed using archived formalin-fixed paraffin-embedded (FFPE) tissue samples derived from 125 OSCC patients treated by surgery between January 1996 and November 2007 at the Hospital Universitario Central de Asturias, Spain. FFPE tissue specimens were sourced from the Principado de Asturias BioBank (PT17/0015/0023) and processed following standard operating procedures. Clinicopathologic data were collected from clinical records. Written informed consent was obtained from all patients. The inclusion criteria for all the participants enrolled were: (i) primary OSCC (International Classification of Disease-10 diagnosis codes: C02.0, C02.1, C02.2, C02.3, C03.0, C03.1, C04.0, C04.1, C05.0, C06.0, C06.1 and C06.2), (ii) treatment-naïve patients from whom we had formalin-fixed paraffin-embedded (FFPE) primary biopsy tissue samples and (iii) with a minimum follow-up of at least three years for alive patients. In addition, the exclusion criteria were: (i) OSCC with immediate postoperative death, (ii) recurrent disease, (iii) neoadjuvant chemo- or radiotherapy and (iv) missing survival data. The tumor histological grade was determined according to the World Health Organization classification [41] and the clinical staging was assessed according to the eighth edition of the AJCC [42] classification.

All patients underwent surgery of the primary tumor with curative intention as well as neck dissection without receiving any preoperative treatment with radiotherapy and/or chemotherapy. All research procedures have been performed in accordance with the World Medical Association Declaration of Helsinki.

### 2.2. Immunohistochemistry (IHC)

OSCC samples were processed for tissue microarray (TMA) as previously described [10,11,43]. Infiltration of TIL subtypes as well as PD-L1 expression were evaluated in the total selected cohort of 125 OSCC patients. However, immunohistochemical analysis of some CSC markers could not be evaluated in the whole sample due to the lack of representative tumor samples available in the OSCC TMAs. The following primary antibodies were used: mouse monoclonal anti-CD4 (Dako, clone 4B12, 1:80 dilution), mouse monoclonal anti-CD8 (Dako, clone C8/144B, prediluted), rabbit monoclonal anti-FoxP3 (Cell Signalling Technology, clone D6O8R, 1:100 dilution), mouse monoclonal PD-L1 antibody (clone 22C3, PD-L1 IHC 22C3 pharmDx, Dako SK006; 1:200 dilution), NANOG (D73G4) XP® rabbit monoclonal antibody (Cell Signaling technology, Inc.; 1:200 dilution), anti-SOX2 rabbit polyclonal antibody (AB5603, Merck Millipore; 1:1000 dilution), OCT4 (clone MRQ-10, Roche; prediluted), Nestin (clone 10C2, Invitrogen; 1:200 dilution) and IgG anti-podoplanin monoclonal antibody (clone D2-40, Covance Inc. formerly Signet Catalog No. 730-01; 1:100 dilution) using the Dako EnVision Flex+ Visualization System (Dako Autostainer) and diaminobenzidine chromogen as the substrate. Negative controls were prepared by omitting the primary antibody. Positive controls were prepared using appropriate positive control slides. Tonsil tissue was used as a positive control for CD4, CD8 and FOXP3; PD-1 or placenta for PD-L1; seminoma for NANOG and OCT4; and squamous carcinoma for SOX2, Nestin and PDPN.

The IHC results were independently evaluated by five observers (three pathologists (F.J.S.S., F.D.I. and V.B.L.), J.P.R. and J.M.G.-P.), blinded to clinical data. CD4, CD8 and FOXP3 immunostainings in both the tumor nests and the surrounding stroma were scored using the average of positively stained cells in each 1 mm^2^ area from three independent high-power representative microscopic fields (HPFs, 400×; 0.0625 μm^2^). The CD8/FOXP3 ratio was calculated using the CD8 and FOXP3 results. Since the cut-off value for TILs has not been standardized, the median of the number of CD4^+^, CD8^+^ and FOXP3^+^ cells/mm^2^ was chosen and used for survival analysis with patients categorized into high (above the median) and low (below the median) subgroups. Semiquantitative scoring was applied for PD-L1 staining, which was considered positive if more than 10% of tumor cells were positively stained, as described previously [43]. NANOG immunostaining was scored as negative or positive, following previously described methods [10,11]. SOX2 staining scores were classified as negative or positive expression on the basis of values below or above the median cut-off value of 10%, respectively. Nestin staining was classified as low or high expression on the basis of values below or above the median cut-off value of 60% of cells with positive membranes and/or cytoplasmic staining. For PDPN assessment, immunostaining intensity was graded on a scale from zero to three (zero = negative, one = weak, two = moderate, three = strong), and in addition, the proportion of PDPN immunoreactive tumor cells was scored as zero (0%), one (1 to 33%), two (34 to 66%) or three (more than 66 % of tumor cells positive) [44]. To grade Podoplanin expression, each tumor was assigned an immunoreactive score (IRS) [44] based on the percentage of Podoplanin-positive tumor cells and the intensity of the staining. The IRS was calculated by multiplying the staining intensity by the percentage of positive cells resulting in a range from zero to nine. IRS equal or less than four was considered low, and the remaining values were seen as high [45].

The density and distribution of TILs in OSCC, CD4, CD8 and FOXP3 expression was analyzed by immunohistochemistry in 125 OSCC samples using tissue microarrays; these figures are shown elsewhere [46] (Figure 1A,B). The mean numbers of CD4^+^ TILs (Figure 1C,D) in the tumor nests and the surrounding stroma were 6.02 ± 12.29 cells per mm^2^ and 54.60 ± 68.15 cells per mm^2^, respectively. The mean CD8^+^ TILs (Figure 1E,F) in the tumor nests and the tumor stroma were 47.88 ± 57.16 and 178.45 ± 203.21 cells per mm^2^, respectively; the mean numbers of FOXP3^+^ lymphocytes (Figure 1G,H) in the tumor nests and the surrounding stroma were 3.11 ± 6.47 per mm^2^ and 15.65 ± 24.54 per mm^2^, respectively (Appendix A).

### 2.3. Statistical Analysis

Data were analyzed using SPSS software version 18 (IBM Co., Armonk, NY, USA). Continuous variables (CD4^+^, CD8^+^ and FOXP3^+^ cells) were expressed as the means ± standard deviation (SD), and absolute and relative frequencies were calculated for categorical variables. Correlations between the number of stromal or tumoral CD4^+^, CD8^+^ and FOXP3^+^ cells were calculated using Spearman’s correlation coefficient. TILs means per mm^2^ for CD4^+^, CD8^+^ and FOXP3^+^ cells were correlated with clinicopathological variables using Student’s *t* test or the Mann-Whitney U test when variables had non-normal distributions. The endpoint of this study was disease-specific survival (DSS), calculated as the time from the date of surgery to death for the tumor. Censoring was defined as loss of follow-up or alive and free of recurrence at the end of the follow-up. Receiver operating characteristic (ROC) and area under the curve (AUC) analysis were used to estimate the predictive value of infiltrating TILs in OSCC patient survival. The survival rates were estimated by the Kaplan-Meier method and compared using the log-rank test. Moreover, the Cox regression model was applied to calculate hazard ratios (HRs) with 95% confidence intervals (95% CI). The factors that were significant in univariate analysis were then analyzed using the multivariate Cox regression model to determine the independent prognostic factors in the presence of other prognostically relevant covariates. All *p*-values were based on the two-sided statistical analysis, and a *p*-value less than 0.05 was considered statistically significant.

## 3. Results

### 3.1. Patient Characteristics

The patients were predominantly men (82 vs 43) and ranged in age from 28 to 91 years (mean 58.69, standard deviation 14.34 years). The tumor subsite was classified as the tongue (*n* = 51, 41%), floor of the mouth (*n* = 37, 30%), gingiva (*n* = 22, 18%), buccal mucosa (*n* = 7, 6%), retromolar area (*n* = 6, 4%) or palate (*n* = 2, 1%). Smoking habit and alcohol consumption were respectively reported in 84 (67%) and 69 (55%) patients. AJCC classification [42] T1 was detected in 27 (22%) patients, T2 in 54 (43%), T3 in 16 (13%) and T4 in 28 (22%) patients. pN0 classification was observed in 76 (61%), pN1 in 25 (20%) and pN2 in 24 (19%) patients. According to overall AJCC disease stages, 20 (16%) had stage I, 32 (26%) had stage II, 26 (20%) had stage III and finally, 47 (38%) had stage IV. Regarding histopathologic degree of differentiation, 80 (64%) OSCCs were well-differentiated, 41 (33%) moderately and 4 (3%) poorly-differentiated. 

### 3.2. Immunohistochemical Evaluation of CD4^+^, CD8^+^ and FOXP3^+^ TILs in OSCC Tissue Specimens

There was a strong positive correlation between stromal and tumoral infiltration of the different TILs subtypes (Figure 1, Appendix A), with the only exceptions being tumoral CD8^+^ and stromal FOXP3^+^, which were not significantly correlated (Appendix A).

### 3.3. Associations between CD4^+^, CD8^+^ and FOXP3^+^ TILs Density and Clinicopathological Variables

All TILs markers (CD4, CD8, FOXP3) were evaluated for the whole series (125 cases). Stromal CD4^+^ TILs infiltration was significantly associated with a non-smoking habit, well-differentiated tumors and tongue tumoral location. On the other hand, tumoral CD4^+^ TILs infiltration was significantly associated with older age, female gender and tobacco or alcohol consumption, as well as tongue location. Stromal CD8^+^ TILs were significantly associated with well-differentiated tumors, and tumoral CD8^+^ TILs were associated with the absence of tobacco or alcohol consumption. Finally, stromal FOXP3^+^ TILs showed a significant association with T classification and with tumors located in the tongue (Table 1).

### 3.4. Associations between CD4^+^, CD8^+^, FOXP3^+^ TILs Density and Expression of CSC Markers

The expression of NANOG and OCT4 was evaluated by immunohistochemistry in 122 cases, SOX2 in 121 cases, Nestin in 93 cases and PDPN in 85 cases in which representative tumor tissue was available. Positive SOX2 and NANOG expression was respectively detected in 49 (40%) and 39 (32%) of the OSCC samples in this series, as we previously reported [10,11]. Overall, infiltration of CD4^+^ and CD8^+^ TILs but not FOXP3^+^ TILs was inversely correlated with the expression of the CSC markers NANOG and SOX2. Concordantly, stromal/tumoral CD4^+^ and CD8^+^ TILs were significantly higher in tumors harboring negative expression of NANOG, except for tumoral CD8^+^ TILs that did not attain significance (Table 2). Noteworthy, the CD8^+^/FOXP3^+^ TILs ratios in both tumor nests and stroma were consistently higher in tumors harboring negative expressions of NANOG and SOX2; however, significant associations were only reached between the tumoral CD8^+^/FOXP3^+^ ratio and both SOX2 and NANOG. None of the tumor samples showed OCT4 expression. Positive Nestin and PDPN expressions were respectively detected in 86 of 93 (92%) and 28 of 85 (33%) OSCC samples in this series. Similar to our observations for NANOG and SOX2, higher infiltrations of CD4^+^ and CD8^+^ TILs, but not FOXP3^+^ TILs, were observed in those tumors harboring low expressions of Nestin and PDPN (Table 2). In addition, stromal/tumoral FOXP3^+^ TILs were significantly higher in tumors with a high PDPN expression. In turn, higher stromal/tumoral FOXP3^+^ TILs densities were associated with tumors harboring low Nestin expression, reaching significance in the case of stromal FOXP3^+^ TILs (Table 2).

### 3.5. Associations between CD4^+^, CD8^+^, FOXP3^+^ TILs Density and PD-L1 Expression

Positive PD-L1 expression in more than 10% of tumor cells was detected in 18 (15%) of our series of OSCC samples, as previously reported [43]. High tumoral CD4^+^ and CD8^+^ -infiltrating TILs were significantly associated with PD-L1-positive tumors (Table 3). The tumoral CD8^+^/FOXP3^+^ TILs ratio was significantly higher in PD-L1-positive tumors, while the stromal CD8^+^/FOXP3^+^ ratio was higher in PD-L1-negative tumors (Table 3). 

### 3.6. Impact of CD4^+^, CD8^+^ and FOXP3^+^ Infiltrating TILs on the Survival of OSCC Patients

Follow-up information was available for 125 OSCC patients, ranging from 6 to 230 months with a mean of 74.10 (SD: 57.08) and a median of 61 months. At the end of this study, 17 (14%) patients were lost during the follow-up period, 53 (44%) patients were alive and free of recurrence and finally, 51 (42%) of them died of cancer or showed a non-treatable recurrence. Stromal/tumoral CD4^+^, CD8^+^ and FOXP3^+^-infiltrating TILs were divided into two categories (high/low density) according to their respective median values. As shown in Appendix A, stromal and tumoral FOXP3^+^ TILs were superior to other TIL markers in terms of determining patient prognosis (AUC = 0.384, 95% CI = 0.285–0.483, *p* = 0.028; and AUC = 0.368, 95% CI = 0.270–0.466, *p* = 0.012; respectively). Contrasting this, tumoral and stromal CD4^+^ and CD8^+^ TILs did not show any significant relationship with patient survival (Table 4, Appendix A). However, high densities of stromal and tumoral FOXP3^+^ TILs were significantly associated with more favorable prognosis (*p* = 0.03 and *p* = 0.03, respectively; Table 4 and Figure 2A,B). The median value of the tumoral CD8^+^/FOXP3^+^ ratio was 11.22 and the median value of the stromal CD8^+^/FOXP3^+^ ratio was 10.98. A low tumoral CD8^+^/FOXP3^+^ ratio was found to significantly associate with a better prognosis (*p* = 0.04; Figure 2C).

We also performed a stratified univariate Kaplan-Meier analysis according to CD4^+^, CD8^+^ and FOXP3^+^ TILs densities (Appendix A). A high stromal CD8^+^ TILs density (above the median) was significantly associated with a better survival in patients with ages lower than 65 years (*p* = 0.01), male gender (*p* = 0.02), tobacco (*p* = 0.02) and alcohol consumption (*p* = 0.02), neck lymph node metastasis (*p* = 0.03), well-differentiated tumors (*p* = 0.03), complementary radiotherapy (*p* = 0.01), negative SOX2 expression (*p* = 0.03) and negative NANOG expression (*p* = 0.04). On the other hand, high stromal FOXP3^+^ TILs density (above the median) was significantly associated with better prognosis in female patients (*p* = 0.03), smokers (*p* = 0.02), drinkers (*p* = 0.02), patients with tumors arisen in the tongue (*p* = 0.03) and negative PD-L1 expression (*p* = 0.02). Similarly, a high tumoral FOXP3^+^ density was associated with increased survival in patients younger than 65 years (*p* = 0.02), female patients (*p* = 0.01), smokers (*p* = 0.01), alcohol drinkers (*p* = 0.01), patients treated with complementary radiotherapy (*p* = 0.04) and tumors with negative PD-L1 expression (*p* = 0.01). A low stromal CD8^+^/FOXP3^+^ ratio was significantly associated with a better prognosis in patients with female gender (*p* = 0.009), history of alcohol consumption (*p* = 0.03), tongue tumor location (*p* = 0.01) and well-differentiated tumors (*p* = 0.03) (Appendix A).

Multivariate analysis further revealed that clinical stage, which combines pT and pN classification, and a high stromal CD8^+^/FOXP3^+^ ratio were the only parameters independently associated with DSS (HR = 2.195, *p* = 0.01 and HR = 2.039, *p* = 0.03, respectively) (Appendix A).

## 4. Discussion

In this study, we assessed the relationship between CD4^+^, CD8^+^ and FOXP3^+^ TILs, as well as with PD-L1 and CSC markers in OSCC, and their association with clinicopathologic features and disease prognosis. We found a strong positive correlation among stromal and tumoral CD4^+^, CD8^+^ and FOXP3^+^ cell numbers, with the only exceptions being tumoral CD8 and stromal FOXP3. These findings support the presence of different TILs subtypes in OSCC TME interacting with one another to exert their effects, which highlights the idea that the numbers and functions of TILs are dynamic in nature. The relationship between cancer cells and host immune cells is very complex. The immune system promotes the elimination of tumor cells, but this system plays a dual role in tumor evolution, and cancer cells can display an escape from the immune system, maintaining the tumor progression [47]. In this study, the numbers of CD4^+^ and CD8^+^ lymphocytes were not significant predictors of the patient’s survival in the multivariate analysis, results that are consistent with other reports [3,37,48,49,50,51]. Of interest, here, CD4^+^ and CD8^+^ cells showed higher numbers in patients who did not have tobacco or alcohol habits; these associations suggest that tobacco and alcohol consumption not only play a carcinogenic role in OSCC but may also influence the host’s immune system response. Studies showed that CD8^+^ cells are a critical barrier to the initial development of tumors [52]. CD8^+^ TIL infiltration of tumors has been largely related to favorable outcomes in different cancers and to a favorable response to chemoradiotherapy [53,54], while immunosuppressive Tregs can confer good or poor prognosis depending on the context. Some evidence has been provided that higher numbers of Tregs are linked to worse prognosis in HNSCCs [55], but others have noticed conflicting results [56]. We herein identified stromal and tumoral FOXP3^+^ TILs as independent prognostic factors in the whole OSCC series, and all the studied lymphocyte subtypes also showed prognostic relevance in different strata of other variables. Our results indicate a major impact of FOXP3^+^ Tregs, more so than CD8^+^ T cells and CD4^+^ T helpers, on the patient’s outcome. Furthermore, in a multivariate analysis, the stromal CD8^+^/FOXP3^+^ ratio was revealed as an independent prognostic indicator together with the clinical stage of the disease. Immunohistochemistry allows for the location of TILs in the tumor nests or tumor stroma separately, and this could be of a paramount importance because in previous reports, stromal TILs seem more relevant than intra-epithelial ones [57]. 

OSCC is a cancer exposed to a microbial environment within the oral cavity, and it is often ulcerated and heavily infiltrated by TILs. Troiano et al. [58] classified oral tongue squamous cell carcinomas into three groups: immune-inflamed, when TILs were found inside the tumor mass in proximity to tumor cells; immune-excluded, when TILs were located into the stroma, outside the tumor; and finally, immune-desert, when tumors lacked TILs. Tumors of the latter group showed the worse prognosis among tongue carcinomas. It is not entirely understood how higher Treg infiltrates could be beneficial for patient survival [59]; however, we can speculate that this phenomenon may be due to the relationship between Tregs and an inflamed TME [60]. In this regard, various immunosuppressive markers, such as PD-L1 and FOXP3, have been associated with T-cell-inflamed tumors and better prognosis [61]. On the other hand, FOXP3 is essential to functional maintenance of Tregs, whose infiltration in the TME negatively regulates the immune response against tumors, favoring tumor growth and consequently related to a poor prognosis in HNC [62]. However, we found here a paradoxical relationship between high FOXP3^+^-infiltrating TILs and good prognosis. A meta-analysis comprising 15,512 patients demonstrated that the prognostic role of FOXP3 was highly influenced by the tumor site and tumor stage [38]. Thus, when analyzing the impact of immunological parameters on OSCC patient survival, we identified that a low density of both stromal and tumoral FOXP3^+^ lymphocytes showed a poorer outcome compared with their counterparts with high FOXP3^+^ lymphocyte numbers. Thus, high FOXP3^+^ cell infiltration was associated with poor prognosis in the majority of solid tumors and no prognostic effect was observed in ovarian and pancreatic cancers, whereas in others, such as head and neck cancers, tumor infiltrating FOXP3^+^ T cells were associated with favorable prognosis [33,38]. Our results are in line with some authors [3,24,36,59,63], but contrasted to a few others [24,50,64]. Hence, the prognostic relevance of FOXP3^+^ Tregs is not completely understood. Furthermore, it is necessary to take in account that FOXP3^+^ T cells are highly heterogeneous with respect to their genotype and phenotype, having been subdivided into three functional subpopulations: effector, resting and non-suppressive cytokine-secreting Tregs [65]. Here a high density of FOXP3^+^ T cells was associated with extended survival in patients in the whole sample, but especially in patients treated not only by surgery but also by radiotherapy. Interestingly, in patients who underwent surgery alone, only a modest and non-significant improvement in DSS was noticed. Radiotherapy affects the TME, and it has been observed that is able to enhance the antitumor immune response in rectal and pancreatic cancer [66,67]. It has been suggested that the composition of the TME before treatment is of less importance than the antitumor immune response induced by the radiotherapy [68]. In this sense, we studied the composition of TME in OSCC tissue specimens obtained from primary surgical treatment, i.e., pre-radiotherapy TME, and still a relationship with a better prognosis was observed. The conflicting results from different studies suggest that FOXP3^+^ T cells act differently in different anatomical subsites and tumor stages [69], changing to pro-tumorigenic cells as the tumor progresses, or that it might depend on their origin, with only FOXP3^+^ T cells recruited from the peripheral blood involved in tumor elimination [57]. Together with the aforementioned functional heterogeneity of FOXP3^+^ subpopulations, a possible explanation for the striking association of Tregs with a favorable prognosis could be related to a role in suppressing or at least down-regulating inflammatory reactions that promote tumor progression by growth factor and cytokine production by immune cells, killing macrophages and monocytes [33,38,70]. Importantly, we have identified various subsets of patients based on clinicopathological factors and TILs subtypes that aid in establishing different outcomes. Nevertheless, some caution is warranted because of possible over fitting of the data as well as power loss in the data analysis due to the large number of variables analyzed and the small size in some clinical groups. The anatomical location of the tumor within the oral cavity may influence the prognostic role of T cells. Oral cancer arises from diverse subsites, and tongue carcinomas should be studied separately as they have unique epidemiological characteristics different from those of other oral cavity cancers [69]. The present study reported that stromal and tumoral CD4^+^ T cell densities and also stromal FOXP3^+^ T cell density were higher in tumors located in the tongue in comparison to tumors arisen in other sites within the oral cavity. These results seem to be partially consistent with findings provided by Kashima et al., [71] who found a predominance of CD8^+^ and FOXP3^+^ T cells in tongue squamous cell carcinomas. In this study, a high stromal FOXP3^+^ TILs density and a low stromal CD8^+^/FOXP3^+^ ratio were associated with a better outcome in tongue carcinomas, while CD4^+^ T cell density did not show any prognostic relevance, possibly due to the diversity of T-helper subtypes and hence their functional complexity. Here, TILs’ density was separately evaluated in two different compartments within OSCC TME. Using a similar procedure, Naito et al. [72] found that CD8^+^ T cells infiltrating the tumor islands affected prognosis positively in colorectal cancer, while CD8^+^ T cells located in the tumor stroma had no effect on prognosis. Conversely, Menon et al. [73] showed the prognostic relevance of stromal CD8^+^ TILs in colorectal cancer. 

In addition, we also found a positive association between PD-L1 expression and a higher CD4^+^ and CD8^+^ TILs densities within the tumor islands, which is in accordance with previous studies [68]. Even though CD8^+^ TILs’ density did not globally show a significant impact on the prognosis in our cohort of OSCC patients, it is worth mentioning that a higher TILs density was associated with a better prognosis in specific subsets of OSCC patients with neck node metastasis, well-differentiated tumors and patients treated with surgery and radiotherapy. It has been shown that PD-L1 correlates with the presence of CD8^+^ TILs in the TME, [74] and PD-L1 can induce apoptosis in activated antigen-specific CD8^+^ cells [75]. A negative correlation between PD-L1^+^ and CD8^+^ cells in OSCC has been observed, which implies that the blockade of PD-L1 induces local immune activation [76]. However, we found a significant positive association between CD8^+^ TILs in the tumor compartment and positive PD-L1 expression, which could reflect the presence of an adaptive immune resistance mechanism triggered by CD8^+^ T cells that secretes IFNγ, suggesting that CD8^+^ TILs may have a function other than a cytotoxic one [77]. Thus, if T cells are exposed to a persisting antigen for two to four weeks, T-cell exhaustion may be established [48]. OSCC comprises a heterogeneous cell population, including CSCs, with tumor-associated antigens present, which can be potentially targeted by immune cells [59]. CSCs may escape from the host’s immunosurveillance by producing cytokines in the TME and paralyzing the immune system responses, by converting a subset of immature myeloid DCs into TGF-β-secreting cells, thus attracting Tregs to tumors, facilitating spreading and metastasis [74,75,77,78,79] and inhibiting CD8^+^ T cells [14]. CSCs may express different stem-like markers such as SOX2, NANOG and OCT4 that allow them to or preclude them from adapting to challenging environmental situations. Thus, various CSs subpopulations can co-exist within a tumor and expand according to their own hierarchy, driving tumor evolution [8]. Notably, in this study, CD4^+^ and CD8^+^ TILs were inversely and significantly associated with the expression of two important CSC-related regulators NANOG and SOX2, and analogous associations were observed for two additional CSC markers, Nestin and PDPN, with the only exceptions being tumoral CD8^+^ and Nestin. Moreover, the tumoral CD8^+^/FOXP3^+^ ratio was inversely and significantly associated with the expression of NANOG and SOX2, thereby suggesting an inverse association between cytotoxic and helper TILs infiltration and stemness in OSCC. A similar trend was found for PDPN but not Nestin. NANOG not only contributes to the regulation of pluripotency in stratified epithelia [80]; it also mediates tumor cell proliferation, epithelial-mesenchymal transition and escape from immune system [78]. NANOG expression in OSCC is regulated by SOX2, OCT4, KLF4, AGR2, NOTCH1 and miR-34a [13]. However, OCT4 expression was not detected in any tumor in our OSCC cohort. Concordantly, analogous observation was obtained from the analysis of a series of 348 tumors located in oropharynx, hypopharynx and larynx [81], while quite remarkably, different OCT4 antibodies were employed in both studies. Ghazi et al. [82] found that SOX2 expression did not correlate with OCT4 expression in OSCC. Additionally, Vijayakumar et al. [83] studied 20 cases of OSCC and 20 cases of oral epithelial dysplasia (OED), and found that SOX2 expression was higher in OSCC than in OED; most cases predominantly showed high SOX2 expression accompanied by negative OCT4 expression. Together these data highlight that SOX2 expression in HNSCC and OSCC seems to primarily be independent of OCT4 transcriptional control. OCT4 and SOX2 have been reported to play different roles in CSC biology. In the absence of OCT4 expression, neoplasms could not be initiated from normal tissues, but without SOX2 expression, the neoplastic cells could not be self-renewed to maintain tumor growth [82]. Nestin is considered a CSC marker in epithelial neoplasms and has been shown to play key roles in differentiation, proliferation, migration, invasion, metastasis and survival of malignant neoplastic cells through regulation of the cytoskeleton and progenitor cells [83]. In this study, Nestin expression was present mainly in the cytoplasm of tumor cells and showed an inverse relationship with the stromal/tumoral CD8^+^, CD4^+^ and FOXP3^+^ cell densities and CD8^+^/FOXP3^+^ ratios similar to other CSC markers studied (i.e., NANOG and SOX2), although the only significant association reached was with stromal FOXP3^+^ TILs. In contrast, PDPN expression, a demonstrated CSC marker in squamous carcinoma cells [21], while showing similar relationships with different TILs subtypes such as SOX2 and NANOG, was only significantly correlated with stromal and tumoral FOXP3^+^ cells. As far as we know, our study is the first to consistently show striking inverse correlations between CD8^+^ TIL, CD4^+^ TIL and CD8^+^/FOXP3^+^ ratios in OSCC TME and the expression of various well-established CSC markers such as NANOG, SOX2 and Nestin, and also between FOXP3^+^ lymphocytes and PDPN expression. According to these findings, we hypothesize that cytotoxic and helper TILs’ infiltration does not seem to be related to CSC niche maintenance, but opposed to stemness maintenance and CSCs’ escape from the immune system. These findings are in line with our previous study demonstrating an inverse association between both stromal and tumoral M2 macrophages and NANOG expression [84]. Our results also pose a possible link between PDPN expression and immunosuppression in OSCC. The literature on the prognostic significance of TILs is heterogeneous in terms of sample sizes, patient cohorts and methodological differences among studies mainly due to the diverse techniques, scoring methods and cut-off values used, which may altogether contribute to contradictory results [45]. Moreover, immunohistochemically phenotyping T lymphocyte subsets is limited by the fact that this technique does not provide any information on their functionality [54].

## 5. Conclusions

These findings demonstrate that stromal/tumoral FOXP3^+^ TILs infiltration and a low stromal CD8^+^/FOXP3^+^ ratio are significantly associated with improved survival, and the stromal CD8^+^/FOXP3^+^ ratio emerges as a significant independent prognostic factor in OSCC. Moreover, high CD4^+^ and CD8^+^ T infiltration and a high tumoral CD8^+^/FOXP3^+^ ratio are significantly associated with high tumoral PD-L1. Strikingly, this study uncovers an inverse relationship between high infiltration of CD8^+^ and CD4^+^TILs with the absence/low expression of various CSC markers (i.e. NANOG, SOX2 and Nestin), while FOXP3^+^ Treg infiltration is significantly correlated with PDPN expression. Further research is warranted to validate these results.

## Figures and Tables

**Figure 1 biomedicines-09-00653-f001:**
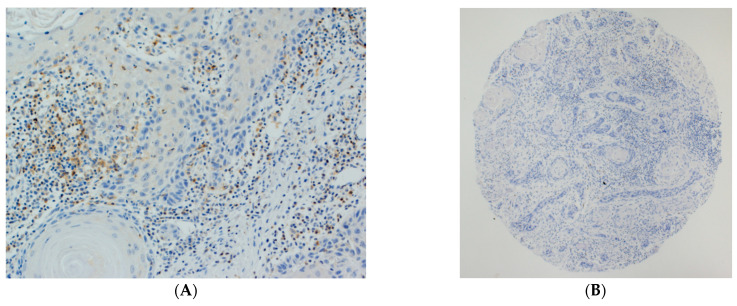
Representative examples of oral squamous cell carcinomas (OSCC) showing high (**A**) and low (**B**) densities of TILs. Stromal and tumoral staining of CD4^+^ (**C**,**D**), CD8^+^ (**E**,**F**) and FOXP3^+^ (**G**,**H**) TILs. Magnification: 50× (**B**); 100× (**A**,**C**,**E**); 200× (**D**,**F**,**G**); 400× (**H**).

**Figure 2 biomedicines-09-00653-f002:**
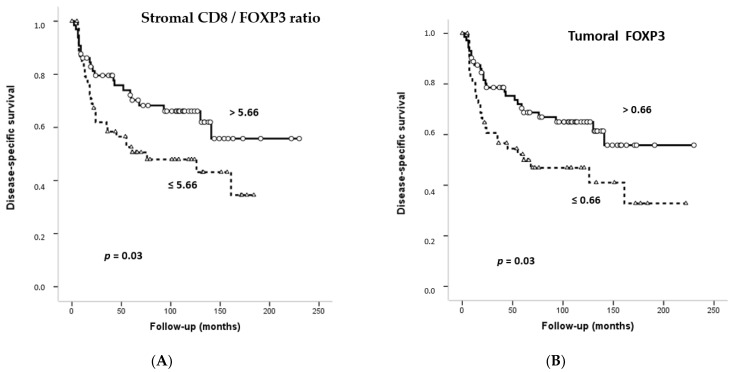
Kaplan-Meier disease-specific survival curves in the cohort of 125 OSCC patients categorized by stromal FOXP3^+^ (**A**), tumoral FOXP3^+^ (**B**) or stromal CD8^+^/FOXP3^+^ ratio (**C**). Median values were used as cut-off points. *p* values were estimated using the log-rank test.

**Table 1 biomedicines-09-00653-t001:** Associations between stromal and tumoral CD4, CD8 and FOXP3 TILs and the clinicopathological parameters in the cohort of 125 OSCC patients.

Variable	Number	Stromal CD4 Mean (SD)	*p*	Tumoral CD4 Mean (SD)	*p*	Stromal CD8 Mean (SD)	*p*	Tumoral CD8 Mean (SD)	*p*	Stromal FOXP3 Mean (SD)	*p*	Tumoral FOXP3 Mean (SD)	*p*
Age (years)													
<65	77	43.87 (42.35)	0.05	3.51 (5.43)	0.01	158.24 (160.38)	0.16	43.47 (58.60)	0.28	18.32 (27.33)	0.13	3.79 (7.70)	0.09
≥65	48	71.82 (94.14)		10.01 (17.94)		210.88 (256.15)		54.85 (54.68)		11.46 (18.92)		2.05 (3.67)	
Gender													
Female	43	66.37 (96.18)	0.25	7.92 (13.85)	0.03 *	205.31 (276.68)	0.37	51.88 (67.82)	0.57	16.19 (22.44)	0.85	3.40 (6.00)	0.71
Male	82	48.43 (46.92)		5.02 (11.34)		164.37 (151.46)		45.75 (50.94)		15.35 (25.73)		2.95 (6.74)	
Tobacco													
No	41	76.18 (96.80)	0.03 *	10.26 (18.58)	0.01 *	215.76 (226.61)	0.18	67.25 (69.83)	0.02	14.65 (22.50)	0.75	2.76 (4.95)	0.67
Yes	84	44.07 (45.62)		3.93 (6.70)		160.25 (189.54)		47.34 (5.19)		16.15 (25.62)		3.28 (7.13)	
Alcohol consumption													
No	56	67.10 (86.31)	0.08	8.54 (16.18)	0.004 *	208.14 (222.55)	0.14	58.69 (64.46)	0.04 *	13.84 (20.70)	0.45	2.38 (4.40)	0.25
Yes	69	44.46 (47.02)		3.95 (7.27)		154.36 (184.20)		38.98 (49.08)		17.15 (27.40)		3.72 (7.78)	
T classification													
T1 + 2	81	55.93 (54.83)	0.76	6.28 (12.52)	0.75	192.29 (206.21)	0.3	45.69 (58.84)	0.56	17.96 (26.50)	0.02 *	3.55 (7.14)	0.3
T3 + 4	44	52.15 (88.27)		5.56 (11.99)		152.99 (197.33)		51.85 (54.40)		11.34 (19.97)		2.30 (4.96)	
N classification													
N0	76	52.74 (50.56)	0.7	5.22 (8.31)	0.43	176.77 (196.70)	0.9	50.73 (59.57)	0.49	14.24 (21.01)	0.42	2.78 (4.83)	0.48
N+	49	57.48 (89.39)		7.25 (16.68)		181.07 (214.97)		43.52 (53.56)		17.84 (29.32)		3.62 (8.46)	
Stage													
I + II	52	57.77 (55.79)	0.66	5.93 (9.88)	0.94	197.14 (224.40)	0.38	47.67 (63.87)	0.97	16.93 (24.08)	0.62	3.43 (5.61)	0.64
III + IV	73	52.34 (76.04)		6.09 (13.80)		165.14 (187.14)		48.02 (52.42)		14.74 (24.98)		2.88 (7.05)	
Grade													
Well	80	63.23 (80.43)	0.02	6.09 (11.39)	0.94	204.61 (235.63)	0.02	48.23 (59.11)	0.92	14.70 (23.95)	0.57	3.15 (6.98)	0.92
Moderate + Poor	45	39.26 (33.17)		5.91 (13.86)		131.96 (115.05)		47.25 (54.20)		17.28 (25.72)		3.04 (5.56)	
Site													
Tongue	51	73.87 (90.52)	0.01 *	9.52 (17.06)	0.02	200.96 (229.44)	0.3	43.11 (50.32)	0.44	21.67 (32.63)	0.04	4.26 (8.85)	0.15
Others	74	41.32 (42.89)		3.58 (6.43)		162.95 (183.01)		51.21 (61.61)		11.52 (15.92)		2.33 (4.01)	
Recurrence													
No	71	47.50 (49.79)	0.25	4.78 (8.86)	0.23	171.85 (173.57)	0.67	45.65 (55.24)	0.62	18.85 (27.78)	0.06	3.61 (7.26)	0.05
Yes	54	62.63 (86.47)		7.64 (15.61)		187.13 (238.10)		50.76 (59.96)		11.55 (19.12)		2.47 (5.30)	
Second primary tumor													
No	106	53.13 (70.30)	0.57	5.72 (12.24)	0.52	170.75 (205.11)	0.31	44.95 (51.89)	0.18	15.99 (25.91)	0.71	3.09 (6.64)	0.92
Yes	19	62.79 (55.56)		7.70 (12.80)		221.45 (191.76)		64.07 (80.24)		13.76 (15.36)		3.24 (5.65)	

Tumoral but not stromal CD8^+^ infiltration was significantly associated with non-smokers (*p* = 0.02) and non-alcohol drinkers (*p* = 0.04), and stromal CD8^+^ infiltration was significantly higher in well-differentiated tumors (*p* = 0.02). Higher stromal, but not tumoral FOXP3^+^ TILs infiltration was significantly associated with smaller tumors (T1 and T2) (*p* = 0.02), as well as tumors arisen in the tongue (*p* = 0.04) (Table 1). It was not possible to calculate the CD8^+^ /FOXP3^+^ ratio in all the cases since for several of them, the TIL density was zero. In fact, the stromal CD8^+^/FOXP3^+^ ratio was determined in 106 cases, and the tumoral CD8^+^/FOXP3^+^ ratio in 80 cases. Higher stromal and tumoral CD8^+^/FOXP3^+^ ratios were significantly found in older patients (*p* = 0.001), and a higher tumoral CD8^+^/FOXP3^+^ ratio was significantly associated with non-smokers (*p* = 0.01) and non-alcohol drinkers (*p* = 0.005). No other significant associations were observed with any of the remaining clinicopathological variables (Appendix A). ******* U Mann-Whitney test.

**Table 2 biomedicines-09-00653-t002:** Associations between tumoral and stromal CD4^+^, CD8^+^ and FOXP3^+^ TILs and expression of CSCs markers.

Factor	SOX2		*p*	NANOG		*p*	Nestin		*p*	PDPN (IRS)		*p*
	Negative	Positive		Negative	Positive		Low	High		Low	High	
Stromal CD4 (mean, SD)	63.66 (81.02)	43.61 (43.92)	0.11	63.00 78.30)	36.71 (35.64)	0.01	58.28 (53.67)	49.61 (52.18)	0.30	61.45 (88.09)	47.78 (36.98)	0.61
Tumoral CD4 (mean, SD)	6.89 (14.65)	5.06 (8.13)	0.43	7.51 (14.32)	3.29 (6.12)	0.03	6.26 (6.83)	5.15 (11.36)	0.21	8.02 (16.48)	4.63 (7.09)	0.96
Stromal CD8 (mean, SD)	215.23 (234.37)	128.21 (142.87)	0.01	200.83 (229.44)	134.07 (131.46)	0.04	229.90 (172.05)	163.05 (187.08)	0.13	189.45 (239.22)	186.60 (161.68)	0.45
Tumoral CD8 (mean, SD)	53.44 (58.72)	41.95 (55.66)	0.04	53.95 (61.19)	36.11 (48.35)	0.11	39.28 (67.17)	45.52 (58.46)	0.55	51.36 (63.00)	45.38 (42.81)	0.90
Stromal FOXP3 (mean, SD)	15.57 (22.17)	16.48 (28.32)	0.84	14.81 (24.46)	17.05 (24.92)	0.64	32.85 (37.47)	13.48 (20.90)	0.01	9.75 (17.19)	20.02 (21.76)	0.003
Tumoral FOXP3 (mean, SD)	2.54 (4.60)	4.12 (8.57)	0.24	3.22 (7.37)	2.90 (4.36)	0.80	7.54 (9.36)	2.25 (4.39)	0.05	2.10 (4.20)	3.29 (5.53)	0.03
Stromal CD8/FOXP3 ratio (mean, SD)	56.48 (124.57)	38.03 (78.71)	0.15	62.08 (129.78)	25.05 (36.24)	0.64	14.15 (10.93)	52.28 (127.31)	0.54	71.66 (137.10)	48.51 (113.58)	0.08
Tumoral CD8/FOXP3 ratio (mean, SD)	35.32 (56.90)	22.39 (57.74)	0.008	39.29 (67.86)	12.58 (15.63)	0.02	7.42 (6.04)	35.28 (68.76)	0.35	46.52 (84.19)	29.58 (35.36)	0.96

**Table 3 biomedicines-09-00653-t003:** Association between CD4^+^, CD8^+^ and FOXP3^+^TILs and PD-L1 expression (clone 22C3) in the cohort of 125 OSCC patients.

Factor	Tumor PD-L1		*p*
	≤10%	>10%	
Stromal CD4 (mean, SD)	50.09 (48.50)	86.40 (134.59)	0.27
Tumoral CD4 (mean, SD)	4.57 (7.89)	15.16 (24.77)	0.01
Stromal CD8 (mean, SD)	161.87 (158.97)	285.01 (365.01)	0.17
Tumoral CD8 (mean, SD)	40.10 (49.76)	98.49 (73.77)	0.004
Stromal FOXP3 (mean, SD)	16.18 (25.41)	14.57 (21.23)	0.80
Tumoral FOXP3 (mean, SD)	3.22 (6.86)	2.98 (4.43)	0.88
Stromal CD8/FOXP3 ratio (mean, SD)	50.03 (116.61)	43.89 (49.56)	0.08
Tumoral CD8/FOXP3 ratio (mean, SD)	26.63 (48.44)	49.02 (92.22)	0.01

**Table 4 biomedicines-09-00653-t004:** Kaplan-Meier and univariate Cox cancer-free survival analysis in the cohort of 125 patients with OSCC.

Parameter	Number	Censored Patients (%)	Cancer-Free Survival Time (95% CI)	*p*	Hazard Ratio	95% CI
pT classification						
T1–T2	81	53 (65)	151.82 (129.03–174.99)	0.001	2.49	1.44–4.30
T3–T4	44	19 (43)	77.62 (54.19–101.04)			
pN classification						
N0	76	49 (65)	127.96 (109.43–146.49)	0.01	1.92	1.12–3.31
N+	49	23 (4)	108.58 (77.88–139.28)			
Stage						
I + II	52	36 (69)	140.09 (120.15–160.04)	0.002	2.4	1.33–4.32
III + IV	73	36 (49)	113.33 (87.73–138.93)			
Grade						
Well	80	44 (55)	127.85 (103.73–151.98)	0.59	0.85	0.48–1.52
Moderate + Poor	45	28 (62)	121.63 (96.25–147.01)			
Stromal CD4^+^						
≤32.66	62	34 (55)	127.86 (102.28–153.45)	0.6	0.86	0.50–1.49
>32.66	63	38 (60)	137.49 (110.54–164.44)			
Tumoral CD4^+^						
≤2.66	59	31 (53)	120.39 (93.85–146.93)	0.42	0.8	0.47–1.38
>2.66	66	41 (62)	142.01 (115.42–168.60)			
Stromal CD8^+^						
≤118	63	32 (51)	109.62 (81.85–137.39)	0.05	0.58	0.33–1.01
>118	62	40 (65)	152.28 (126.67–177.90)			
Tumoral CD8^+^						
≤24.65	61	35 (57)	133.16 (105.84–160.47)	0.95	1.01	0.59–1.74
>24.65	64	37 (58)	111.28 (90.91–131.66)			
Stromal FOXP3^+^						
≤5.66	60	29 (48)	96.14 (75.24–117.05)	0.03	0.56	0.32–0.98
>5.66	65	43 (66)	152.79 (126.93–178.66)			
Tumoral FOXP3^+^						
≤0.66	53	25 (47)	105.47 (77.32–133.62)	0.03	0.54	0.31–0.93
>0.66	72	47 (65)	151.97 (127.55–176.38)			
Stromal CD8/FOXP3^+^ ratio						
≤10.9853	43	30 (70)	162.89 (132.67–193.11)	0.04	1.9	0.99–3.64
>10.9853	63	32 (51)	97.82 (78.78–116.87)			
Tumoral CD8/FOXP3^+^ ratio						
≤11.2254	39	28 (72)	163.64 (131.68–195.59)	0.06	2.01	0.95–4.27
>11.2254	41	22 (54)	100.67 (77.71–123.63)			

*p* values were estimated using the log-rank test.

## Data Availability

The data presented in this study are available within the article and Appendix A.

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
