# Peer review of "Prognostic Relevance of CD4+, CD8+ and FOXP3+ TILs in Oral Squamous Cell Carcinoma and Correlations with PD-L1 and Cancer Stem Cell Markers"

_biomedicines, 2021, doi:10.3390/biomedicines9060653_

Round 1
Reviewer 1 Report
the corrections made have improved the paper, so it is possible to publish it in the current form
Author Response
Reviewer 1:
The corrections made have improved the paper, so it is possible to publish it in the current form.
Response: We thank the reviewer for his/her positive comments considering that our work has been considerably and adequately improved for publication in the current form.
Reviewer 2 Report
The research subject is interesting, being about oral cancer, but not sufficient
The presentation is poor. The study is about "patients treated by surgery between January 1996 and November 2007". Is missing a long time period of research. The references are found in disorders in the text. The manuscript is not in the journal template.
Author Response
Reviewer 2:
Point 1: The research subject is interesting, being about oral cancer, but not sufficient
Response 1: It is difficult for us to understand this comment/opinion regarding the research subject, which the reviewer considers interesting because it is about oral cancer although not sufficient. Obviously, we cannot agree with this statement and find quite inappropriate not to specify any reason or argument to support the reviewer’s view. In addition, the reviewer has not raised any specific concerns and/or recommendations on how we could improve our manuscript; therefore we cannot provide any additional information or data to adequately address this issue.
Point 2: The presentation is poor. The study is about "patients treated by surgery between January 1996 and November 2007". Is missing a long time period of research. The references are found in disorders in the text. The manuscript is not in the journal template.
Response 2: Similar to our previous response, the reviewer states that the presentation is poor; however, he/she does not provide any specific recommendations/suggestions on how to improve it. The reviewer also indicates that "patients treated by surgery between January 1996 and November 2007". Is missing a long time period of research”. Certainly, the patients enrolled in this study were surgically treated between 1996 and 2007; however, we must clarify that after the treatment all the patients were followed up until 2020. Therefore, there is no missing time on this research; on the contrary patients were adequately followed for a long period of years to assess the impact on patient survival. References have been checked and we have indeed verified that all of them appear in correct order in the text. Finally, the reviewer points that “The manuscript is not in the journal template”. Honestly, we must say that we have strictly followed indications from the journal and, as the Editor knows well, this version of the manuscript compiling text and Figures/Tables was actually generated by Biomedicines Editorial Team and according to the MDPI policies.
Reviewer 3 Report
In this manuscript the Authors evaluated the tumour-infiltrating lymphocytes (TILs) and the immunohistochemical expression of cancer stem cell (CSC) markers and PD-L1 in surgical specimens of oral squamous cell carcinoma (OSCC). This retrospective study investigated the correlations between TILs expression and clinicopathological data and the prognostic role of several biomarkers.
General considerations: The manuscript should be major revised. The first issue concerns its length. The topics should be summarized, focusing on the study aims and on the literature data. The English grammatical and lexical form should be revised (example: Line n° 69-74, 87-88, 124-133, 136-138, 153, 155, 163, 167, 171, 240, 328, 334, 348, 366, 371). The sentences are excessively long, making difficult their clear understanding. The phrase structures should be reformulated, the verbal tenses and forms should be corrected, and the several grammatical repetitions should be reviewed (Line n°50-54, 56, 82-85, 91-97, 124-133, 136-154, 248-290, 305-309, 314-335, 351-369, 373-377, 406-432, 480-511, 543-557). The abbreviations should be defined at their first mention.
Title: The title does not adequately express the study aims.
Introduction: The cancer stem cell markers used in the study have been explained except for NANOG. The repetitive and unclear periods should be reformulated (see the above-mentioned lines).
Results: the results should be clearer and more concise. Only the most significant results should be reported since the data can be consulted in the Tables. In the “2.1 Section” clarify the abbreviation “S1” (Line 97). It is essentially specifying how many samples of the case series (125 OSCC) undergone to the different immunohistochemical evaluations. Especially, it is should specify the exclusion criteria of some samples from the several analyses. Line 151 reported “in the remaining samples no tumor was detected”. The case series should not include only histological samples containing tumor tissue? Furthermore, this specific regarding only two biomarkers. The line 203-204 reported subjective consideration that should not be expressed in this section. The horizontal string of Table 2 and Table 3 should be corrected. It is declared to possess the follow-up data of 121 patients. However, in the Table 4 it is reporting a cohort of 125 patients. The number of cases included in the survival analysis and the exclusion criteria should be specified.
Discussion: the discussion lacks contextual connections. Moreover, it should be extensively synthesized, removing repetitive concepts and reformulating unclear periods (see the above-mentioned lines). Moreover, as the importance of the topic, I recommend updating the literature regarding the prognostic role of TILs in patients affected by OSCC (for your convenience, doi: 10.1002/cam4.3440).
Materials and methods: lines 434-455 reported clinicopathological data that should be explained on results section or into a Table. The inclusion and exclusion criteria should be clarified. Furthermore, the reference used to subclassify the tumor site, the reference of grading system, the follow-up period, and the control (positive and negative) tissues should be explained. Have you obtained the informed consent from each patient? It is not declared in the Manuscript. Moreover, how many pathologists evaluated the immunohistochemical sections? The immunostaining score system (lines 480-511) should be summarized. Why you also consider the presence of non-treatable recurrence to define the DSS?
Conclusion: this section should be more concise.
Author Response
Reviewer 3:
In this manuscript the Authors evaluated the tumour-infiltrating lymphocytes (TILs) and the immunohistochemical expression of cancer stem cell (CSC) markers and PD-L1 in surgical specimens of oral squamous cell carcinoma (OSCC). This retrospective study investigated the correlations between TILs expression and clinicopathological data and the prognostic role of several biomarkers.
Response: We thank the reviewer for his/her meticulous revision and for all the valuable recommendations.
Point 1: General considerations: The manuscript should be major revised. The first issue concerns its length. The topics should be summarized, focusing on the study aims and on the literature data. The English grammatical and lexical form should be revised (example: Line n° 69-74, 87-88, 124-133, 136-138, 153, 155, 163, 167, 171, 240, 328, 334, 348, 366, 371). The sentences are excessively long, making difficult their clear understanding. The phrase structures should be reformulated, the verbal tenses and forms should be corrected, and the several grammatical repetitions should be reviewed (Line n°50-54, 56, 82-85, 91-97, 124-133, 136-154, 248-290, 305-309, 314-335, 351-369, 373-377, 406-432, 480-511, 543-557). The abbreviations should be defined at their first mention.
Response 1: Manuscript length has been shortened, and the language issues carefully reviewed and amended to improve clarity. However, we want to point out that due to the number of biomarkers assessed in this study (CD4+, CD8+, FOXP3+, PD-L1, NANOG, SOX2, OCT4, Nestin, Podoplanin), it has been difficult to reduce further the text while still maintaining sufficient description of the results/data and to avoid removing any relevant information.
Point 2: Title: The title does not adequately express the study aims.
Response 2: The title has been modified according to the reviewer’s suggestion.
Point 3: Introduction: The cancer stem cell markers used in the study have been explained except for NANOG. The repetitive and unclear periods should be reformulated (see the above-mentioned lines).
Response 3: Information about NANOG has been now included in the Introduction.
Point 4: Results: the results should be clearer and more concise. Only the most significant results should be reported since the data can be consulted in the Tables. In the “2.1 Section” clarify the abbreviation “S1” (Line 97). It is essentially specifying how many samples of the case series (125 OSCC) undergone to the different immunohistochemical evaluations. Especially, it is should specify the exclusion criteria of some samples from the several analyses. Line 151 reported “in the remaining samples no tumor was detected”. The case series should not include only histological samples containing tumor tissue? Furthermore, this specific regarding only two biomarkers. The line 203-204 reported subjective consideration that should not be expressed in this section. The horizontal string of Table 2 and Table 3 should be corrected. It is declared to possess the follow-up data of 121 patients. However, in the Table 4 it is reporting a cohort of 125 patients. The number of cases included in the survival analysis and the exclusion criteria should be specified.
Response 4: Results section is now more concise and hopefully more clearly presented. The abbreviation S1 corresponds to Supplementary Figure S1, which is now clearly indicated.
Regarding the number of samples evaluated by IHC, this study comprises 125 patients and the corresponding OSCC tissue samples. However, there was no enough representative tumor tissue from these samples in the TMAs to study all the biomarkers. Precisely, CD4, CD8, FOXP3, and PD-L1 were studied in 125 cases, while NANOG and OCT4 were studied in 122 cases, SOX2 in 121, Nestin in 93, and PDPN in 85 cases. In a previous revision of this manuscript, following the reviewers’ comments, the number of CSCs markers studied was extended from the original two NANOG and SOX2 to also include OCT4, Nestin and PDPN, with the remaining tissue material available.
On the other hand, following the reviewer’s recommendation, further changes have been included in this new version as listed below:
- The subjective consideration of lines 203-204 has been deleted.
- The horizontal strings of Tables 2 and 3 have been modified.
- The number of cases included in the survival analysis was 125.
- Exclusion criteria have been specified in Material and Methods
Point 5: Discussion: the discussion lacks contextual connections. Moreover, it should be extensively synthesized, removing repetitive concepts and reformulating unclear periods (see the above-mentioned lines). Moreover, as the importance of the topic, I recommend updating the literature regarding the prognostic role of TILs in patients affected by OSCC (for your convenience, doi: 10.1002/cam4.3440).
Response 5: Discussion has been modified and references updated, as suggested.
Point 6: Materials and methods: lines 434-455 reported clinicopathological data that should be explained on results section or into a Table. The inclusion and exclusion criteria should be clarified. Furthermore, the reference used to subclassify the tumor site, the reference of grading system, the follow-up period, and the control (positive and negative) tissues should be explained. Have you obtained the informed consent from each patient? It is not declared in the Manuscript. Moreover, how many pathologists evaluated the immunohistochemical sections? The immunostaining score system (lines 480-511) should be summarized. Why you also consider the presence of non-treatable recurrence to define the DSS?
Response 6: The following changes have been made to this new version of the manuscript, in agreement to the reviewer’s recommendation,
- Clinicopathological data included in lines 434-455 have been moved to the Results
- Inclusion and exclusion criteria are now specified in Material and Methods
- The references used for tumor classification and grading have been added.
- Information concerning patient consent is now mentioned in Material and Methods
- IHC has been evaluated by five authors. Three of them are highly experienced pathologists.
- Immunostaining scoring is now summarized.
- Non-treatable recurrence has been considered in the definition of DSS because in the absence of treatment in these patients, the death is very close. However, the exact date of death is taken into account in the survival analysis.
Point 7: Conclusion: this section should be more concise.
Response 7: Conclusion has been made more concise.
Round 2
Reviewer 3 Report
The manuscript is suitable for publication.
This manuscript is a resubmission of an earlier submission. The following is a list of the peer review reports and author responses from that submission.
Round 1
Reviewer 1 Report
The manuscript entitled “Prognostic relevance of CD8+/FoxP3+ TIL ratio in oral squamous 2 cell carcinoma and inverse correlation with the cancer stem cell 3 markers NANOG and SOX2” which is contributed by Lequerica-Fernández et al. provided a similar manuscript from their previous research from Cancers (Cancers 2020, 12, 1764; doi:10.3390/cancers12071764) with different immune cell markers.
Honestly, I cannot discriminate the novelty and importance between these two papers.
Major comments:
The figure 1 legend cannot match to picture. For example, Magnification of A and B are the same but the pictures are not.
In figure 2, please label clear in different groups.
Reviewer 2 Report
This research titled “Prognostic relevance of CD8+/FoxP3+ TIL ratio in oral squamous cell carcinoma and inverse correlation with the cancer stem cell 3 markers NANOG and SOX2”, discuss the pivotal role of TILs in oral squamous cell carcinoma, by immunohistochemistry assay on tissue microarray. The authors analyzed the involvement of CD4+, CD8+,FOXP3+ and CD8+/FOXP3+ in the disease progression and patients’ outcome. Further, they study a possible correlation with PD-L1 and cancer stem cell markers.
This study in not completely innovative, other authors have already addressed the study of immunological TME in oral squamous cell carcinoma. The cancer stem cell part adds something new, but it opens other limits. However, the authors approached the experimental part with precision and method and the discussion is well structured.
Major points:
- The TMA use is a limit in the study of microenvironment: have the authors validated the observation on same full section? In alternative, E&E of TMA should be showed (also as supplementary) to demonstrate the validity of the test sample (areas really representative for the presence of Tils).
- The introduction is too long: it should be reduced.
- Only two stem markers are not sufficient to give certainty of stemness. At least two additional markers should be added (OCT4; Nestin, CD166 or CD133).
Minor points
- the choice of cut-off should be better explained?
- the part about TMA use lines 480-491 is well known and it is not necessary.
- the list of references should be update